# Left Frontal EEG Power Responds to Stock Price Changes in a Simulated Asset Bubble Market

**DOI:** 10.3390/brainsci11060670

**Published:** 2021-05-21

**Authors:** Filip-Mihai Toma, Makoto Miyakoshi

**Affiliations:** 1NEXARCH Lab, Bucharest 010374, Romania; mihai.toma@nexarchlab.com or; 2Swartz Center for Computational Neuroscience, Institute for Neural Computation, University of California San Diego, 9500 Gilman Drive, La Jolla, CA 92093-0559, USA

**Keywords:** neuroeconomics, financial bubble, financial decision-making, EEG, stimulus preceding negativity, default mode network

## Abstract

Financial bubbles are a result of aggregate irrational behavior and cannot be explained by standard economic pricing theory. Research in neuroeconomics can improve our understanding of their causes. We conducted an experiment in which 28 healthy subjects traded in a simulated market bubble, while scalp EEG was recorded using a low-cost, BCI-friendly desktop device with 14 electrodes. Independent component (IC) analysis was performed to decompose brain signals and the obtained scalp topography was used to cluster the ICs. We computed single-trial time-frequency power relative to the onset of stock price display and estimated the correlation between EEG power and stock price across trials using a general linear model. We found that delta band (1–4 Hz) EEG power within the left frontal region negatively correlated with the trial-by-trial stock prices including the financial bubble. We interpreted the result as stimulus-preceding negativity (SPN) occurring as a dis-inhibition of the resting state network. We conclude that the combination between the desktop-BCI-friendly EEG, the simulated financial bubble and advanced signal processing and statistical approaches could successfully identify the neural correlate of the financial bubble. We add to the neuroeconomics literature a complementary EEG neurometric as a bubble predictor, which can further be explored in future decision-making experiments.

## 1. Introduction

Crashes of financial bubbles have had severe negative impacts on society. Bubbles are deviations from the market equilibrium for which standard asset pricing theory does not apply, and they form when prices are significantly increased above a fundamental intrinsic value [1,2]. Their causes could be attributed to over-inflated expectations of market participants towards future prices, which in turn are a consequence of cognitive biases such as aggregate overconfidence [1] or herding behavior [3]. These psychological factors, what Keynes deemed as “animal spirits”, drive deviations from market equilibrium, warranting supplementary methods from the field of experimental cognitive neuroscience to understand bubble dynamics in addition to the traditional econometric approaches using historical data [4]. Fortunately, there is a surge of interest in the brain–computer interface (BCI) using low-cost, low-density, desk-top or even wearable EEG devices rather than expensive and immobile neuroimaging machines with superconductive or radioactive elements. As most stock trades occur in front of computer monitors today, the distance between non-invasive cognitive neuroscience and stock trading has never been closer. The use of low-cost desktop/wearable EEG-based BCI devices may become a standard practice in the near future for stock trading to support the user’s decisions or to calibrate behavior and provide real time neurofeedback in this sense [5].

The primary limitation of low-cost, low-density EEG systems is given by the small number of scalp electrodes. The solution is discussed in the reference paper of the *keyhole hypothesis* [6]. The name is taken from the analogy that seeing inside a room from a keyhole only provides narrow field of view, but if the observer knows the room well (e.g., his/her own room), one can still understand the ongoing situation in the room. Similarly, a measurement from low-dimensional EEG provides a narrow field of view, but we can still obtain valuable insights if it is evaluated based on the prior knowledge of the observed phenomenon. Fortunately, there are neuroeconomic studies which applied neuroscientific research methods to economics to understand the cognitive processes underlying financial decision making [7,8]. Financial decision-making tasks during bubbles mainly recruit the frontal cortex, and the list of repeatedly reported frontal regions include the medial prefrontal cortex (PFC) [9,10,11,12], lateral PFC [13,14,15], and the orbitofrontal cortex (OFC) [16,17]. Particularly, the lateral frontal cortex is responsible for value-based choice, ranging from consumption-based decisions [18,19], intertemporal choice [20] to deductive reasoning [21]. Furthermore, in terms of risk-taking, several studies used the popular Iowa Gambling Task [22] in understanding brain dynamics, which have isolated FRN and P300 components in the centro-frontal region [23,24]. These studies provide useful prior information that the frontal lobe is the key region in financial decision making. Moreover, the theory of the generative mechanism of the scalp-recorded EEG predicts that a dominant EEG contributor is likely to be found on the surface of the neocortex, i.e., a continuum of crowns of the neocortical gyri [25]. Taken together, we predict that the lateral regions of the frontal cortex make preferable targets for the current scalp EEG measurement. However, as far as we know, trial-by-trial EEG analysis on financial bubbles using realistic inexpensive BCI-friendly desk-top environments has not been reported in the literature.

To demonstrate a proof of concept, we designed an EEG study on a financial decision-making task in a financial bubble market using an inexpensive BCI-friendly desk-top environment, with an Emotiv EPOC headset. Details on EGG-based BCI data acquisition and analysis using this headset are detailed by Paszkiel and Szpulak [26]. We developed an unconventional EEG analysis approach using mass-univariate linear modeling [27] on independent component analysis (ICA)-decomposed EEG power changes in the time-frequency domain [28,29,30] to examine trial-by-trial changes toward the financial bubble. We set our region of interest (ROI) in the lateral frontal regions based on the prior information suggested from the literature of neuroimaging studies on neuroeconomics. The goal of the study is to demonstrate that the trial-by-trial EEG power change reflects the stock price in the lateral frontal regions as predicted.

## 2. Materials and Methods

### 2.1. Subjects

A total of 28 healthy adults (10 women) participated in the study. EEG was successfully recorded from 26 subjects. There were two sessions, totaling 52 datasets, but 3 of them were not usable due to technical error. Thus, 49 EEG datasets were used for the final analysis. Written informed consent was obtained from all participants. The protocol of the experiment was approved by THE Q-AGENCY, the company from which the EEG headset was rented. Full consent was given in accordance with the Helsinki accord by each participant. The participants were both (i) last year Bachelor’s and Master’s students in economics and (ii) professionals with economic background studies working in the financial industry. We also included three former traders with investing experience. The rationale behind the selection was to obtain an as homogenous group as possible with general financial markets knowledge. Thus, proper comparison and analysis of data between the two experiments could be carried out relevantly within the context of the research question.

### 2.2. Electroencephalography (EEG) Recording

Scalp EEG recordings were obtained using an Emotiv EPOC headset (May–June 2017, Bucharest, Romania) using 14 NaCl wet electrodes placed to the locations of AF3, AF4, F3, F4, F7, F8, FC5, FC6, T7, T8, P7, P8, O1 and O2, according to International 10–20 system. EEG signal was recorded with a sampling rate at 128 Hz. Skin–electrode impedance was decreased by using saline liquid until the level required by the Emotiv Pro software was reached and turned green, signifying a impedance below 20 kΩ. Data from 3 subjects could not be analyzed due to excessive noise.

### 2.3. Experimental Design

The experiment is a replication of [31], adapted using EEG. The task was repeated once to observe changes after having experienced the bubble crash once. The experimental software was designed with the help of two developers, and it supplied market prices which subjects could not impact by means of supply and demand. Thus, the market was simulated and while participants thought they were actively trading, their actions did not impact the market price. This approach provides advantages both from the perspective of it being similar to real-world trading (as it is unlikely for individual traders to impact prices using small orders) and the fact that the experiment can be run using one subject at a time, thereby ensuring simplicity. A similar approach was taken by [12], for which [5] identify similar upsides. Figure 1A displays market dynamics.

We shortly review the experiment design, turning the reader to the original study for more details. Each subject starts with six stocks and 100 units of experimental cash currency. When a stock is bought (sold), the paid price is deducted (added) from the cash at that period. Only one stock can be traded for each period. Each asset produces income for the subject. The stock produces a dividend of either 1 or 0.4 units with equal chances, dividends being independent in time. Cash produces fixed income at 5% for each period. At the end of the experiment, each stock held is sold at a price of 14 units. Any price above 14 is considered a bubble. All information was the same for all subjects. Prior to the experiment, subjects read detailed experimental instructions and took a quick quiz in order for the experimenter to assess their understanding. Almost all participants obtained maximum scores.

The experiment was run twice—each trial run consisted of 30 trading periods—the same for all subjects. Experimental sessions are referred to as Time 1 and Time 2 in the following sections. Each trading period consisted of four different information screens: 1. Holdings, where subjects saw information on the current stock price and cash held, 2. Decision-making, where subjects had to decide on whether to buy, hold or sell one unit of the stock, 3. Outcome, where the decision from the previous screen was re-iterated and 4. Income, where subjects read how much they won. Each information screen lasted for 5 s maximum (decision times differed among subjects), with a 2.5 s fixation cross in between each of them, for a maximum total of 15 min/experiment. Figure 1B provides the window screens that were shown to subjects during the experiment.

For participating in the experiment, each subject received a show-up fee of approximately EUR 4 (in domestic currency using the official exchange-rate at the time) and a variable incentive, in order to induce the necessary risk aversion mandated in order to better simulate market performance. After trading, the final individual earnings were converted into real money (RON) using a rate of 100 cash = 1.5 RON. Each subject won on average EUR 10. Post-experimental questionnaires were filled in by subjects to obtain socio-demographic information. Overconfidence was also measured using a question that captured the better-than-average effect. Simply stated, subjects were asked to rank their position in terms of payoff as opposed to other subjects. Subsequently, the difference between their actual position and what they stated was the metric to deem the subject (non)-overconfident.

### 2.4. Preprocessing of the Scalp-Recorded EEG Data

Figure 2 left shows the flow chart of the preprocessing pipeline for individual data. Offline, the raw data sets were imported to EEGLAB 14.1.2 [32] running under Matlab 2017b (The MathWorks, Inc., Natick, MA, USA) and in-house developed custom code. The continuous data were high-pass filtered at a cutoff frequency of 0.5 Hz (FIR, Hamming window, transition bandwidth 0.5 Hz). Electrode locations in MNI coordinate system were imported. EEGLAB plugin *clean_rawdata()* was applied to remove and reconstruct artifact subspaces using the algorithm artifact subspace reconstruction (ASR) [33,34,35,36,37,38,39]. The plugin *clean_rawdata()* has a function that if the data interpolation by ASR is insufficient in more than 25% of the electrodes within the 0.5-s sliding window, it rejects the window. This approach is capable of removing non-stationary high-amplitude artifacts effectively, which makes an ideal preprocessing stage for subsequent blind source separation, namely independent component analysis (ICA). As a result from cleaning the data, average 1.2 electrodes (SD 1.5, range 0–6) and 4.8% (SD 7.6, range 0–47.9) of data points were rejected. Some of the datasets that showed the large amount of data rejection were excluded in the later stage from the final analysis as a quality control. Adaptive Mixture Independent Component Analysis (AMICA) was applied to the entire time series data decompose multivariate scalp electrode signals into temporally maximally independent components (ICs) [28,29,30,40,41].

After ICA, the rejected data windows by *clean_rawdata()* were recovered by applying symmetric padding to both ends that used a mirrored signal. This preprocess is to recover continuous data for the subsequent wavelet transform. The process of detecting and rejecting the ‘bad electrode’ was included in this process, which was based on the spatial correlation of the time-series data. For more detail, see [38,39]. Morse wavelet (γ = 3, known as ‘airy’) was applied using Matlab Wavelet Toolbox to obtain the time-frequency decomposition of IC activations. The 62 frequency bins logarithmically distributed from 0.9 to 56.1 Hz. The obtained scalogram was log-converted using 10 × log10 and epoched to −2 to +2 s relative to onset of Holdings event, i.e., showing the updated price of the stock to obtain a single-trial ensemble of the event-related spectral perturbation (ERSP). At this point, the dimensions of the obtained data for each subject were 14 (ICs) × 62 (frequencies) × 512 (time points) × 30 (epochs). Similarly, the baseline value, that was defined as the mean power from 0 to 2 s relative to baseline event averaged across all events, was calculated. Then, the baseline values were subtracted from ERSP.

### 2.5. Linear Regression Analysis with Stock-Price Changes on IC ERSP

The obtained single-trial ERSP was uniformly subsampled in the time-frequency dimensions to 14 (ICs) × 31 (frequencies) × 64 (time points) × 30 (epochs). The epochs that contain rejected window period by *clean_rawdata()* were replaced with NaN. Finally, we applied a linear mixed effect (LME) model analysis using the time-series data of the stock price changes for each pixel of 14 (ICs) × 31 (frequency) × 64 (time points) × 49 (datasets). The *t*-statistics obtained from the LME analysis were stored for the final group-level analysis. At this point, datasets with <20 trials were removed to control the quality of the data. As a result, 5 datasets were removed. The final datasets showed average 27.1 (SD 3.0, range 20–30) trials of Holdings available.

### 2.6. Group-Level IC Clustering for the Final T-Test for Time1 and Time 2

Scalp topographies defined by values in columns mixing matrix obtained by ICA were collected from all datasets and submitted for *k*-means clustering. The Silhouette Index ([42]) and Davis–Boldin methods ([43]) both suggested 18 to be the optimum cluster number, while the Calinski–Harabasz method suggested 16 ([44]). We determined to generate 18 IC clusters. Note that these IC clusters were generated based on the spatial information of EEG, which allows one to perform statistical analysis on EEG time-series data without causing double dipping ([45,46]). For each IC cluster, *t*-statistics obtained from the regression analysis in the time-frequency domain were submitted to one-sample *t*-test for each pixel to obtain the final group-level *t*-statistics and *p*-values for Time 1 and Time 2 separately. We set the electrode of interest at FC5; thus, we investigated the IC cluster that captured FC5 most dominantly.

## 3. Results

### 3.1. Behavioral Data

During the trading sessions, subjects earned on average 512 CASH units during the first experimental run and 580 CASH units during the second experiment. The difference reached statistical significance; mean Time 1 = 891 (SD 137, range 478–1189) vs. mean Time 2 = 1032 (SD 109, range 745–1198), t(50) = −4.1, *p* < 0.0001. The result indicated that the participants took advantage of the information obtained from the first trial in repeating the task in the second trial.

### 3.2. EEG Results

Clustering analysis on the scalp topographies of the independent components (ICs) generated 18 IC clusters (Figure 3). One of the IC clusters—Cluster 4—has the peak of the topography in the left frontal regions which overlaps with electrode FC5. Based on our hypothesis that the cognitive processes involved in financial decision-making should be related to the lateral frontal functions, we examined this left frontal IC cluster for further analysis. We also examined Cluster 7 as the nearby area and Clusters 1 and 8 as the corresponding sites in the contralateral hemisphere, but we did not find a significant result to be reported.

Note that the power spectral density (PSD) of typical eye-related IC clusters, such as IC 17 for eye blinks and vertical eye movement and ICs 13 and 16 for horizontal eye movements, is qualitatively different from that of IC4 and other non-eye-artifactual IC clusters in the way that the left-hand slope of the 1/f curve is heavy-tailed. This confirms that eye-related artifacts are successfully decomposed, and ICs clustered in Cluster 4 is generally expected to be independent (i.e., free) of the same kind of artifact.

Linear regression analysis between the single-trial event-related spectral perturbation (ERSP) of IC Cluster 4 and the changes of the stock price was performed for the first (Time 1) and the second (Time 2) experimental sessions separately (Figure 4B). The results revealed that during Time 1, the delta range (1–4 Hz) of the EEG power showed negative correlation with the stock price, which reached statistical significance (Figure 4C). The timing of this ERSP modulation was centered at around latency zero, which is the onset of displaying the updated stock price. When the average was taken across the frequency bins within the time-frequency window of interest (−1 to 1 s and from 2 to 6 Hz), the resulting time-series data showed the negative peak at 62.5 ms (t = −3.062), closely followed at 125 ms (t = −3.024). However, this modulation became much weaker during Time 2, which was not statistically significant. Thus, we confirmed the main effect within Time 1 in this analysis.

## 4. Discussion

The goal of the current study is to test whether the trial-by-trial EEG signal change corresponding to stock price dynamics in a financial bubble can be detected in the lateral frontal regions, as predicted in the neuroimaging literature. We used a BCI-friendly low-cost recording environment and a realistic stock trading task to provide a proof of concept for desk-top neuroeconomics in near future. We applied advanced signal processing and statistical techniques to ask for the neural signature that follows single-trial stock price changes leading up to the financial bubble formation and crash. We found that the left lateral frontal region showed EEG power modulation negatively correlated with the stock prices. Below, we discuss the interpretation of the main finding and its neuroscientific significance.

### 4.1. Latency of the EEG Power Modulation and Relation to Stimulus-Preceding Negativity (SPN)

In the main time-frequency result (Figure 4B), we found a negative correlation with the stock price peaked at 60–120 ms, which is immediately after the onset of the updated stock price on the computer monitor. Obviously, this delta power decrease cannot be the response to the visual stimulus because the latency would be to too early for that. Typically, such visual responses take at least a few hundreds of milliseconds of delay with the earliest around 100 ms poststimulus, which is well-established as the early visual evoked potentials [47]. Given the absence of the expected delay for the visual evoked potential, the observed delta power decrease must be related to predictive functions. We propose that the observed power modulation relates to stimulus-preceding negativity (SPN) in the time-frequency domain, an electrophysiological index of reward anticipation and delivery as well as motivational intensity [48,49]. This is especially likely given the fact that the time interval between the prior gains period and the Holdings period (Figure 1B) was fixed so the onset timing could be predicted exactly. It is known that the higher the amplitude of SPN toward negative polarity, the higher subject’s anticipation to the reward [50,51]. SPN is also larger in tasks where anticipation to fulfillment is envisioned [52]. The underlying generative mechanism of SPN involves the dopaminergic system that is engaged in reinforcement learning [53,54]. As such, SPN has been used as a neuroeconomics paradigm involving monetary rewards as an ERP index. The early example can be found in [55]. If the peak latency of this negative correlation of the delta power observed in the current study is the reflection of anticipation, it makes sense that the negativity peak at 60–120 ms overlaps the latency of the early visual evoked potentials. In other words, this is the latency around which anticipation is replaced with the updated information via visual input, as if the brain experiences ‘the moment of truth’ to verify the expected future. Probably the most well-known stimulus preceding ERP is the contingent negative variation (CNV) [56]. However, SPN is distinguished from CNV on the points that CNV is primarily related to the motor preparation and the maximal potential is observed at the vertex (Cz), while SPN is related to motivation and the maximal potential is observed in the frontal area [49]. Hence, it seems difficult to explain the observed result as CNV. The motivational and reward-dependent aspects of SPN seem to explain the stock price dynamics forming a bubble. It also makes sense that the effect was observed for Time 1 but not for Time 2 as unpredictability was removed, which is in line with the general understanding of the dopaminergic system being related to both motivation and learning [57]. The results from the current study can inform policy makers on how individual decision making translates at an aggregate level such as financial markets [58,59,60].

### 4.2. Polarity of the Modulation and Relation to the Default Mode Network

The results from the present study showed that the stock price change correlated negatively with single-trial delta power change. Just in case, let us make it clear that the negativity of the polarity in SPN as an ERP component does not imply power decrease. EEG power is proportional to the square of EEG amplitude, and polarity of the signal does not matter after taking the square. The task-negative power modulation in the frontal lobe is reported in the literature of default mode network [61,62,63]. Probably, the most well-known cortical nodes of the default mode network are the medial prefrontal cortex (mPFC) and the posterior cingulate cortex (PCC). However, these studies also reported some smaller portions of the lateral frontal also showed corresponding BOLD signal patterns synchronous to that of the mPFC and PCC. We interpret that the negative correlation we found in the present study is related to the dis-inhibition of the default mode network (i.e., reduction in the task-negative network) in the lateral frontal region. In the context of neuroimaging of the default mode network, Nakano and colleagues [64] reported that eye blinking is related to momentary attention release by the activation of the default mode network while deactivating the dorsal attention network [65]. The cortical network associated with eye blinks included some of the lateral frontal regions, similar to the definition of the default mode network. Due to the relatively poor spatial resolution of scalp EEG measurement, it is impossible for us to determine whether the corresponding cortical source of the delta power decrease spatially overlaps with the lateral frontal regions reported [64]. The question about the detail of the localization requires future investigation with a high-density EEG recording system. Another supportive evidence to this view is that in a recent study using EEG and fMRI, it was reported that a sub-anesthetic dose of ketamine caused a reduction in vigilance in behavior and increase in EEG delta power within the lateral frontal regions, which was associated with a reduction in the frontal connectivity [66]. If the reduction in vigilance and the corresponding delta power increase within the lateral frontal region can be interpreted as the activation of the default mode network introduced by ketamine, the reduction in the delta power within the same region may indicate the dis-inhibition of the resting state network and may cause an increase in vigilance. The neuroeconomics paradigm of the boom-and-bust task used in the current study is likely to be representative of increased arousal and vigilance. Taken together, we conclude that the observed negative correlation between the single-trial EEG delta-band power and stock price dynamics may reflect the dis-inhibition of the (task-negative) default mode network that responds to the anticipatory and motivational function that increases arousal in response to the financial bubble.

### 4.3. Limitation

As designed to be a proof-of-concept study, we used an inexpensive wearable EEG recording system to demonstrate the possibility of a current neuroeconomics paradigm in financial decision-making as a future non-laboratory desk-top environment. As such, we could not argue details of spatial location of the estimated ICs. Another limitation is that the quality of the power spectral density of the ICs was not as high as that obtained from standard laboratory data using research-grade EEG recording machines. Due to these limitations, we could not use some of the critical EEG preprocessing approaches, such as automated probabilistic IC labeling solution ([67]). It is desirable to conduct a future study to replicate the current study with more research-oriented set ups with an extended analysis design that includes subgroups of the participants, namely good vs. poor performers, which requires a larger subject sample size. Comparing the two groups with different performances allows one to determine the neural correlates of successful traders, which is one of the key questions in the field of neuroeconomics.

## 5. Conclusions

We conducted a neuroeconomics study on financial bubble dynamics with a stock-trading task and a low-cost BCI-friendly desktop environment. We found that trial-by-trial stock price changes in a simulated asset bubble market were reflected by the delta-band (1–4 Hz) EEG power decrease within the left frontal region. We interpreted the result as stimulus-preceding negativity (SPN) occurring as a dis-inhibition of the lateral frontal resting state network. We conclude that the combination of the financial decision-making task and the inexpensive desk-top BCI-friendly set up was effective in detecting the targeted lateral frontal brain signal expected from the literature of neuroimaging studies. We contribute to the neuroeconomics literature by introducing a complementary EEG measure as a predictor of financial bubbles. At the same time, the methodology and the results can serve as bases for future EEG research on decision making under risk. Hence, we envision future developments of similar studies aiming at cognitive neuro-augmentation. Such studies would then serve as platforms integrating behavioral and brain data with artificial intelligence and machine learning algorithms to better understand decision-making, by calibrating biases and thus optimizing behavior ([68,69]).

## Figures and Tables

**Figure 1 brainsci-11-00670-f001:**
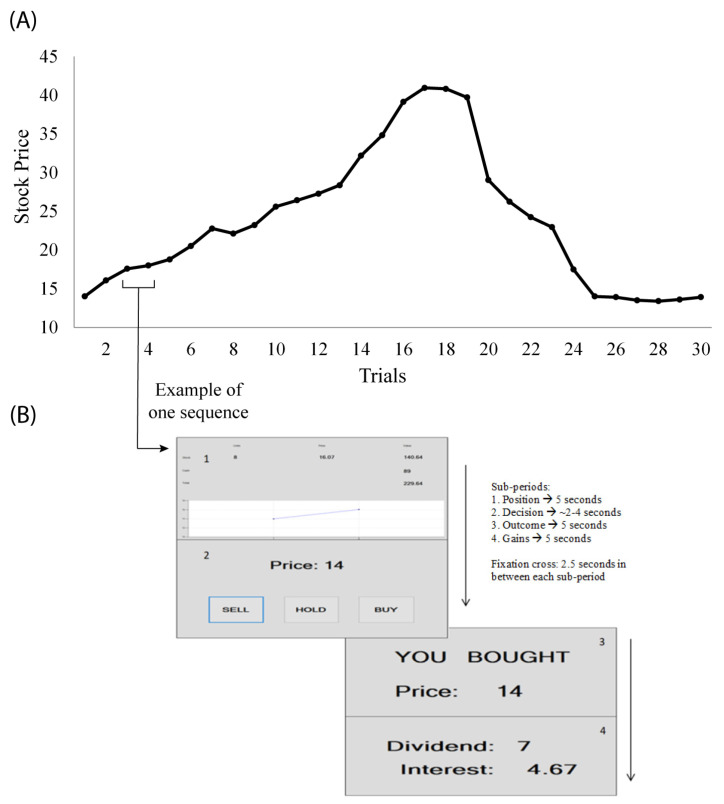
(**A**) The simulated market price for 30 trial periods. All prices above 14 represent a bubble. This time series data were used as a regressor for the subsequent EEG analysis. Prices on the *y*-axis are expressed in experimental currency. (**B**) Images from one trading period subjects saw (numbered in order of appearance): 1. Holdings 2. Decision-making, 3. Outcome and 4. Income (stock dividend and interest from cash). Adapted from Smith et al. (2014) [31]. The main event-related spectral perturbation (ERSP, i.e., EEG power changes caused by behavioral events) reported in this paper is based on single-trial ERSP of +/− 2 s relative to Holdings, i.e., the onset of the updated stock price display.

**Figure 2 brainsci-11-00670-f002:**
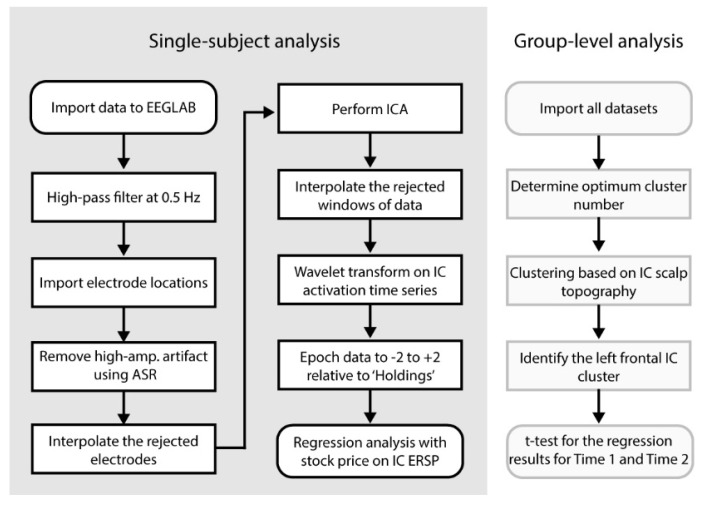
The flow chart of the current data analysis.

**Figure 3 brainsci-11-00670-f003:**
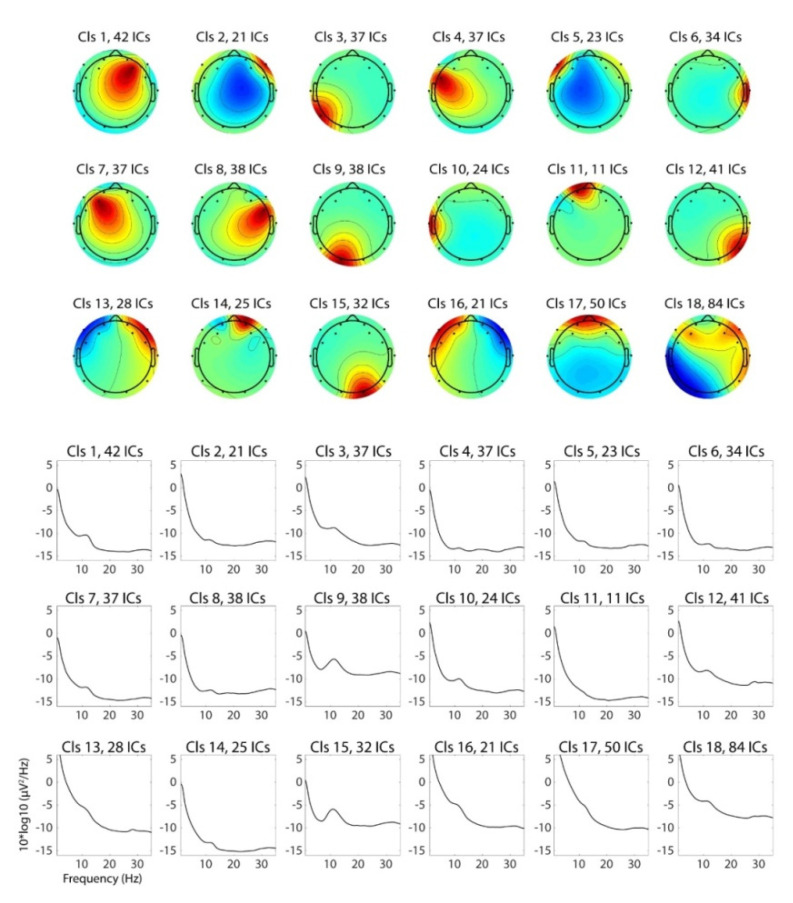
Average of independent-component scalp topographies and power spectral density for each IC cluster. Cluster 4 was the cluster selected for the final analysis. Clusters 1, 7, 8 were examined as well, but did not show significant results.

**Figure 4 brainsci-11-00670-f004:**
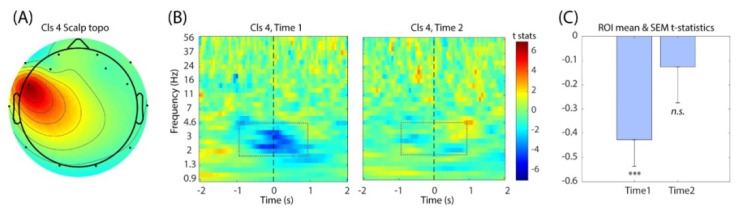
(**A**) The mean IC scalp topography of Cluster 4 that captures the electrode of interest FC5. (**B**) The group-level *t*-test results for Time 1 (left) and Time 2 (right). Note the decrease in the *t*-statistics around time 0 in Time 1, which indicates negative correlation between single-trial EEG power and the stock price. (**C**) The bar graph showing the average *t*-statistics within the time-frequency of window from −1 to 1 s and from 2 to 6 Hz. One-sample *t*-test was performed on Time 1 and Time 2 separately. ***, *p*-Value < 0.005.

## Data Availability

Due to privacy constraints, data are not publicly available and can be provided to researchers upon request for academic purposes.

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
