# Peer review of "Left Frontal EEG Power Responds to Stock Price Changes in a Simulated Asset Bubble Market"

_brainsci, 2021, doi:10.3390/brainsci11060670_

Round 1
Reviewer 1 Report
Dear Authors,
I agree with the authors that financial bubbles are a result of aggregate irrational behavior and cannot be explained by standard economic pricing theory. Authors conducted an experiment in which 28 healthy subjects traded in a simulated market bubble, while scalp EEG was recorded using a low-cost, BCI desktop device with 14 electrodes.
My comments to the study are as follows:
- As part of the Introduction, I propose to extend the background by referring to a greater extent to the methods of acquisition and archiving of brain signals. For example, you can refer to: Methods of Acquisition, Archiving and Biomedical Data Analysis of Brain Functioning, Advances in Intelligent Systems and Computing, Springer from 2018. In addition, I propose to refer to blind EEG separation in the fundamental context of this analysis. For example, you can cite: Characteristics of question of blind source separation using Moore-Penrose pseudoinversion for reconstruction of EEG signal, Advances in Intelligent Systems and Computing, Springer from 2017. This will also keep the bibliography up-to-date.
- I propose to expand the keywords with: BCI, biomedical analysis.
- Instead of the word "helmet", I suggest using: "headset".
- I am also asking for arguments on what basis was the group of participants selected? What criteria did you have in this regard?
- In Fig. 1 there is no axis description in terms of units - please complete.
- Fig. 3 is not made according to the standard for creating Flowcharts, it should be corrected. Data entry should be in a parallelogram. In addition, you should put start and stop on the flowchart.
- Figures (especially: 4, 5) in the article should be enlarged - they are hardly legible.
- Please expand Conclusions with plans for the future.
Reviewer 2 Report
In this paper, the authors undertake the problem of financial bubbles which do not follow the reasonable laws of economics and as we all know lead to disasters of many participating units, under some circumstances to the whole societies. Reading the paper I was impressed by the idea of methodology according to which authors search correlates and anti-correlates between brain cortical activity and tenderness to participation in such a financial bubble. A lot of research has been done in the area of decision making, the most significant seems to be the Iowa Gambling Task by Bechara and Damazio (I suggest referring to them in some), however, herein we have the beginning of quantitative neuroeconomics. Research presented in this paper requires further enrichment and modifications, however, However, the method is scientific, reasonable and novel.
Reviewer 3 Report
The research questions are valid. The methods and study design are appropriate for answering the research question. The Materials and Methods section is sometimes not clearly exposed. However, the study is overall presented and described in terms of contents.
# Materials and Methods
It would be useful to integrate an explanation figure of the various phases of the protocol with the timing of each phase.
Line 136: Why hasn't a Notch filter at 50 Hz been applied? How did you eliminate the instrumental noise? And why did you only apply a high pass filter and not a band pass? Even if the volunteers were seated during the task, you don't have movement and muscle artifacts on the EEG signals? Were you able to eliminate these artifacts with ASR?
Line 144: What criteria was applied to eliminate bad channels?
Line 147: Did you apply the AMICA decomposition for each subject on the entire EEG acquisition?
Line 150 You wrote:” the rejected data windows by clean_rawdata() were recovered by extending the both ends of the windows with reversed signal “. It is not clear this procedure, what did you mean with reverse signal?
Line 152: What are the parameters of the wavelet decomposition? You did not specify.
Line 154: It is not clear which EEG trials were epoched and used for the further ERSP analysis: did you decomposed with wavelet only the epoch from -2 to +2 respect the onset of Holding event?
Line 158-159: What is the baseline event?
Line 162: Why did you sub-sampled? What is the rationale?
Line 165: Do you have a reference for LME linear mixed effect (LME) model analysis?
Line 166: Why 49 datasets?
Fig5: Lake of the capitol letter C close the second time -frequency window
Line 241: there is an additional ‘the’.
